# Altered gating of $K_v$1.4 in the nucleus accumbens suppresses motivation for reward

Bernadette O'Donovan[1], Adewale Adeluyi[2], Erin L Anderson[2], Robert D Cole[1], Jill R Turner[3], Pavel I Ortinski[1]*

[1]Department of Neuroscience, University of Kentucky, Lexington, United States; [2]Department of Drug Discovery and Biomedical Sciences, South Carolina College of Pharmacy, University of South Carolina, Columbia, United States; [3]College of Pharmacy, University of Kentucky, Lexington, United States

**Abstract** Deficient motivation contributes to numerous psychiatric disorders, including withdrawal from drug use, depression, schizophrenia, and others. Nucleus accumbens (NAc) has been implicated in motivated behavior, but it remains unclear whether motivational drive is linked to discrete neurobiological mechanisms within the NAc. To examine this, we profiled cohorts of Sprague-Dawley rats in a test of motivation to consume sucrose. We found that substantial variability in willingness to exert effort for reward was not associated with operant responding under low-effort conditions or stress levels. Instead, effort-based motivation was mirrored by a divergent NAc shell transcriptome with differential regulation at potassium and dopamine signaling genes. Functionally, motivation was inversely related to excitability of NAc principal neurons. Furthermore, neuronal and behavioral outputs associated with low motivation were linked to faster inactivation of a voltage-gated potassium channel, $K_v$1.4. These results raise the prospect of targeting $K_v$1.4 gating in psychiatric conditions associated with motivational dysfunction.
DOI: https://doi.org/10.7554/eLife.47870.001

*For correspondence:
pavel.ortinski@uky.edu

Competing interests: The authors declare that no competing interests exist.

## Introduction

Dysregulated motivation to pursue previously rewarding stimuli is a feature of multiple psychiatric disorders, including depression, schizophrenia, and withdrawal from substance use. Indeed, response to positive motivational situations has been proposed as one of only five behavioral dimensions that link the entire range of psychiatric diagnoses with underlying neurobiological mechanisms (RDoC framework; *Cuthbert, 2015*). Daily fluctuations in motivation for reward is a regular and familiar feature of human experience. However, chronically and persistently low motivation is associated with vulnerability to mental illness, including substance use disorders (*Carroll et al., 2002*; *Janowsky et al., 2003*; *Perry et al., 2007*; *Radke et al., 2015*; *Brennan et al., 2001*). Despite progress in understanding the neurobiology of reward over the last two decades, it remains unclear whether neuronal activity underlying persistent differences in motivation is functionally distinct from a broader spectrum of signaling events mediating reward processing.

The mesolimbic reward circuit is central to the processing of rewarding environmental stimuli. At the center of this circuit is the nucleus accumbens which integrates affective, spatial, and cognitive signals with approach to reward (*Mogenson et al., 1980*; *Di Chiara, 2002*; *Di Chiara et al., 2004*; *Wise, 2004*; *Saddoris et al., 2015*). The principal cells of the nucleus accumbens, striatal projection neurons (SPNs), regulate their firing patterns in a manner that predicts locomotion toward rewards as well as reward omission (*Peoples and West, 1996*; *Nicola et al., 2004a*; *Roitman et al., 2005*; *Wan and Peoples, 2006*). Action potential output of SPNs is modulated by dopamine, a molecule

consistently implicated in behavioral response to motivationally salient stimuli, including action-outcome associations as well as reward avoidance (*Akaike et al., 1987*; *Nicola et al., 2000*; *Ji and Martin, 2012*; *Ortinski et al., 2015*). Dopamine modulates firing of SPNs in the nucleus accumbens via intracellular messengers coupled to the associated G-protein signaling cascades (*Yang et al., 2013*; *Perez et al., 2006*; *Valdés-Baizabal et al., 2015*; *Cantrell et al., 1999*; *Schiffmann et al., 1995*; *Bender et al., 2010*). Among the prominent dopamine targets are the voltage-gated potassium channels, canonical regulators of action potential output.

Voltage-gated potassium channels comprise the most diverse family of ion channels with over a hundred genes coding for potassium channel subunits in addition to multiple channel regulators. These channels are notably capable of suppressing, facilitating, and shaping action potentials and rhythmic activity throughout the central nervous system (*Perez et al., 2006*; *Ji and Martin, 2014*; *Jan and Jan, 1997*; *Kimm et al., 2015*; *Johnston et al., 2010*). For example, in principal neurons of the medial nucleus of the trapezoid body, activation of $K_v1$ channels increases firing threshold and inhibits firing, while activation of $K_v3$ channels accelerates action potential repolarization and promotes high firing rates (*Johnston et al., 2010*). In midbrain dopaminergic neurons, inhibition of $K_v2$ channels increases action potential firing and decreases afterhyperpolarization (AHP), while activation of the large conductance $Ca^{2+}$-activated $K^+$ (BK) channel decreases AHP, but has no effect on action potential firing (*Kimm et al., 2015*). In the striatum, dopamine depletion increases SPN intrinsic excitability and decreases AHP by accelerating the inactivation of the A-type ($I_A$) $K^+$ current (*Azdad et al., 2009*). The kinetic properties of potassium channel gating have been a subject of intense interest over many decades as the timing of channel activation and inactivation affects potassium channel interactions with other ionic conductances to determine membrane excitability and action potential generation.

Recent evidence suggests a prominent role for $K^+$ channels in reward and motivated behaviors (*Han et al., 2013*; *Gelernter et al., 2014*; *Cadet et al., 2017*). For example, several studies indicate that G protein-gated inwardly rectifying $K^+$ (GIRK) channels regulate neuronal firing and behavioral response to addictive drugs (*McCall et al., 2017*; *Rifkin et al., 2017*). In the NAc, where GIRK expression is very low or absent, chronic cocaine treatment has been shown to increase $I_A$ and BK channel currents (*Hu et al., 2004*). Similarly, decreased activation of small-conductance calcium-activated $K^+$ channels in the NAc core has been reported to increase spike output and facilitate motivation to seek alcohol after abstinence (*Hopf et al., 2010*), while microinjection of a $K_v7$ agonist into the nucleus accumbens core has been found to reduce alcohol seeking (*McGuier et al., 2016*). Finally, a genome-wide analysis found rats that compulsively self-administer methamphetamine despite negative consequences (foot shocks) are segregated from non-compulsive methamphetamine takers by both differential RNA expression and DNA hydroxymethylation at a number of genes encoding voltage-gated $K^+$ channels (*Cadet et al., 2017*). Despite this substantial body of evidence that $K_v$ channels play a role in individual responding for drug reward, it remains unclear whether $K_v$-regulation of neuronal excitability may account for variability in responding to naturally reinforcing stimuli.

In this study, we test the hypothesis that individual heterogeneity in effort-based motivation is linked to altered intrinsic excitability of nucleus accumbens shell SPNs. We find that the spectrum of behavioral performance on a classical test of effort-based motivation is mirrored by genome-wide transcriptional differences with a major contribution of voltage-gated $K^+$ channels and dopamine related transcripts. Our electrophysiological analyses provide evidence for a specific voltage-gated $K^+$ channel subtype, $K_v1.4$, as a channel species with potential to specifically target the low motivation phenotype.

## Results

### Individual differences in motivation for sucrose reward

We began by establishing a behavioral profile of motivation for sucrose reward using a PR schedule of reinforcement in seven cohorts totaling 111 rats (*Figure 1*, *Figure 1—figure supplement 1A*). This revealed dramatic variability in PR breakpoints. Across all cohorts, the animals in the top 25% of interquartile distribution (highS rats) reached a mean breakpoint of 270.4 ± 11.3 lever presses, whereas the animals in the bottom 25% of interquartile distribution (lowS rats) reached a mean

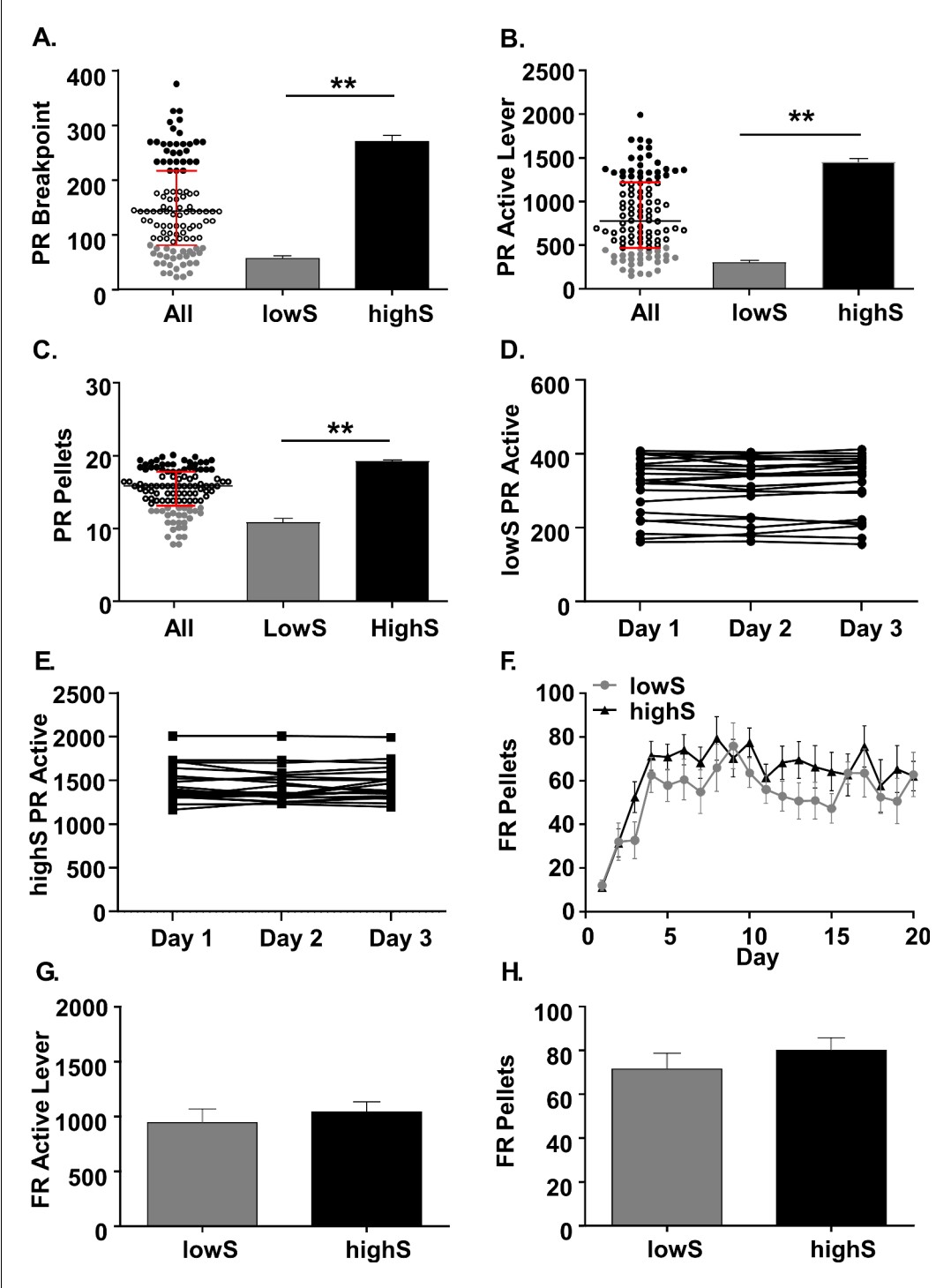

**Figure 1.** Characterization of behavioral variability on a progressive ratio task. (A) *Scatterplot:* Individual breakpoints of rats tested on the PR schedule of reinforcement (N = 111. Black bar: median. Red bars: upper and lower interquartile ranges. *Bar histograms:* Breakpoints were significantly different between rats in the lowest breakpoint quartile (lowS, N = 23) and rats in the highest breakpoint quartile (highS, N = 21). (B) Averaged over the last three days of the PR schedule, the rats in the lowS group pressed the active lever significantly fewer times and (C) earned significantly fewer pellets. (D) Stability of active lever presses in lowS rats over last 3 days on PR schedule. (E) Stability of active lever presses in highS over last 3 days on PR schedule. (F) Rats in the lowS and highS groups acquired the sucrose self-administration task (FR1→FR3→FR10 schedule) at a similar rate. (G) Mean active lever presses over the last three days of FR10 schedule were not different between groups. (H) Mean pellets earned over the last three days of FR10 schedule were not different between groups. **, p<0.01.

DOI: https://doi.org/10.7554/eLife.47870.002

*Figure 1 continued on next page*

Figure 1 continued

The following figure supplement is available for figure 1:

**Figure supplement 1.** Cohort to cohort variability and FR-PR correlation.

DOI: https://doi.org/10.7554/eLife.47870.003

breakpoint of 57.4 ± 4.6 lever presses for a single sucrose reward (*Figure 1A*; $t_{(42)}$=26.35 p<0.0001, unpaired t-test). Consistent with this, the total number of active lever presses per session also differed between highS and lowS rats. HighS rats pressed the active lever an average of 1462 ± 59.2 times, while lowS rats pressed the active lever an average of 296.6 ± 21.2 times per single operant session (*Figure 1B*; $t_{(42)}$=26.49 p<0.0001, unpaired t-test). This difference in operant responding corresponded to an average of nine extra sucrose pellets earned in a single session by highS, relative to lowS, rats (*Figure 1C*; highS: 19.25 ± 0.16 pellets; lowS: 10.9 ± 0.46 pellets, $t_{(42)}$=16.36 p<0.0001, unpaired t-test). To ensure that responding on PR was a stable behavior, animals were assigned to categories following 3 days of stable responding (<10% variability in active lever presses) on the PR schedule (*Figure 1D,E*).

Although classically defined as a test of effort-based motivation, variability in PR performance could reflect differences in ability to learn stimulus-reward associations or locomotor differences in rates of lever responding. To examine this, we analyzed behavioral performance during the fixed-ratio (FR) stage of operant training. The rats that would go on to form lowS and highS group acquired the self-administration task with a similar time course as the FR training progressed from 1 to 3 to 10 lever presses per reward (*Figure 1F*; [Main group effect: $F_{(1,42)}$=2.13, p=0.1] two-way ANOVA). Similarly, the animals did not differ in the number of active lever presses (*Figure 1G*; $t_{(42)}$=0.62, p=0.5442, unpaired t-test) and total pellets earned (*Figure 1H*; $t_{(42)}$=0.93, p=0.36, unpaired t-test) over the final 3 days of FR10 training. There was also no correlation between active lever presses on FR10 and breakpoints achieved on the PR schedule (*Figure 1*, *Figure 1—figure supplement 1B*). This indicates that behavioral variability was specific to the high effort conditions imposed by the PR task. Inactive lever presses were not different between lowS and highS rats on either the FR (lowS: 1.1 ± 0.52; highS: 0.75 ± 0.37; $t_{(42)}$=0.54 p=0.59, unpaired Student's t-test) or the PR (lowS: 2.9 ± 0.6; highS: 3.9 ± 0.7; $t_{(42)}$=1.07 p=0.29, unpaired Student's t-test) schedules of reinforcement suggesting a lack of locomotor differences in approach to lever not associated with reward.

In a subgroup of 16 rats, we also examined whether baseline variability in stress levels may have contributed to PR performance by measuring levels of the stress marker, corticosterone.

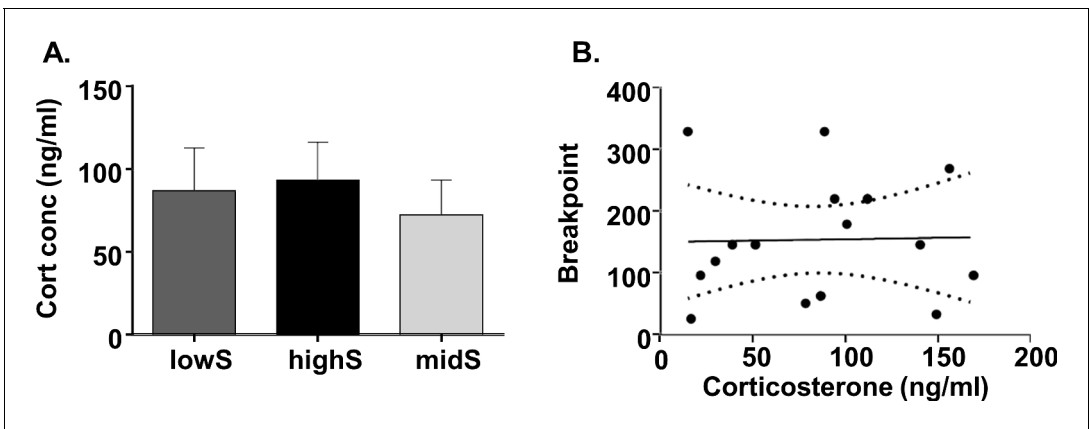

**Figure 2.** Progressive ratio performance does not depend on individual stress levels. (A) Plasma corticosterone levels do not differ between lowS, highS, and intermediate performance (midS) rats. (B) Breakpoint values on the PR schedule do not correlate with plasma corticosterone levels (N = 5/group).

DOI: https://doi.org/10.7554/eLife.47870.004

Corticosterone levels did not differ between rats (*Figure 2A*) and there was no correlation between individual corticosterone levels and PR breakpoints (*Figure 2B*; Pearson's $r^2$ = 0.0005, p=0.93).

Altogether, these data indicate that individual variability in motivation for sucrose is dissociable from performance on low-effort, fixed ratio tasks. Furthermore, variability in motivation that we report is not due to differences in acquisition of stimulus-reward associations, locomotor ability, or baseline differences in HPA axis function.

## Divergent transcriptome profile in highS and lowS rats

Seeking of natural rewards and behavioral performance on progressive ratio schedules of reinforcement critically relies on activity of the NAc shell (*Basso and Kelley, 1999*; *Reynolds and Berridge, 2001*; *Kelley and Swanson, 1997*; *Wirtshafter and Stratford, 2010*). To gain a comprehensive view of molecular drivers of individual variability in motivation for sucrose, we performed genome-wide RNA sequencing of NAc shell tissue. Similar levels of genetic variability could be expected among individuals from a single strain of rats. However, we found that the transcriptome profile diverged between, but not within, groups of rats characterized by their behavioral performance on the PR task. The most pronounced differences appeared between lowS and highS rats, while the transcriptome of animals with intermediate performance on the PR schedule (midS) aligned closer to highS, rather than lowS, rats (*Figure 3A*). Between the lowS and highS groups, a total of 231 transcripts were differentially regulated (*Figure 3*, *Figure 3—source data 1*; $\log_2$fold values $\geq$ 0.5 or$\leq-0.5$). We conducted pathway analysis using Reactome (*Fabregat et al., 2018*; *Milacic et al., 2012*) for a more mechanistic insight into the function of differentially expressed transcripts. This analysis indicated that dopamine and $K^+$ channel-related transcripts accounted for 3 of the top five gene pathways with significant differences in expression between lowS and highS groups. The other two Reactome pathways identified genes associated with extracellular matrix reorganization and cell growth/division (*Figure 3C*). The volcano plot in *Figure 3B* shows differentially enriched genes color-coded in red (upregulated in lowS) and blue (downregulated in lowS) with FDR < 0.05 (horizontal line). Within this pool, we highlight genes known to regulate neuronal activity in the NAc: four genes related to dopamine signaling (*Pdyn*, *Drd1*, *Penk* and *Drd2*), expressed at significantly lower levels in the lowS group, and six genes related to $K^+$ channel signaling (*Hcn4*, *Kcna4*, *Kcnab1*, *Kcnc4*, and *Kcnv1*) that were bi-directionally regulated in the lowS relative to highS group. Overall, RNA sequencing data indicated that behavioral differences in motivation for reward are linked to genomic variability. Particularly prominent were the genes expected to influence NAc neuronal activity and (as in the case of extracellular matrix transcripts) organization of the NAc circuitry.

## Low motivation for sucrose is linked to increased NAc spike output

Our sequencing data indicate a prominent involvement of voltage-gated $K^+$ channels, powerful regulators of neuronal excitability. In vivo studies from other groups suggest an inverse relationship between NAc spiking activity and reward-oriented behavior (*Peoples and West, 1996*; *Nicola et al., 2004a*; *Roitman et al., 2005*; *Wan and Peoples, 2006*) (see Discussion). Therefore, we speculated that low effort-based motivation occurred on a background of increased membrane excitability. Using electrophysiological recordings in brain slices we found that indeed, SPNs of lowS rats fired significantly more action potentials across a range of depolarizing current injections than SPNs of highS rats (*Figure 4A*, [Main group effect: $F_{(1,215)}$=22.52, p=0.01], two-way RM ANOVA). Other measures of excitability, including membrane resistance, resting membrane potential, rheobase, latency to first spike, spike threshold, spike amplitude and spike half-width did not differ between the groups (*Figure 4—figure supplement 1*). However, action potential waveform analysis did show decreased spike afterhyperpolarization in cells from lowS animals (*Figure 4B*; $t_{(16)}$=2.41 p=0.028, unpaired t-test), consistent with a role for voltage-gated $K^+$ channels.

Sequencing data highlighted six $K^+$ channel transcripts differentially expressed in lowS and highS rats, four of which (*Kcnab1*, *Kcna4*, *Kcnc4*, and *Kcnv1*) code for subunits or regulators of $K^+$ channels underlying A-type ($I_A$) currents (*Rettig et al., 1994*; *Tseng-Crank et al., 1990*; *Dallas et al., 2008*; *Hugnot et al., 1996*). We isolated $I_A$ currents using their characteristic property of fast inactivation at depolarized potentials and detected a substantial contribution of these currents to SPN excitability in both lowS and highS groups of animals. Peak $I_A$ amplitude was not significantly different between lowS and highS groups (*Figure 4C*; [Main group effect: $F_{(1,28)}$=2.64, p=0.12], two-way RM

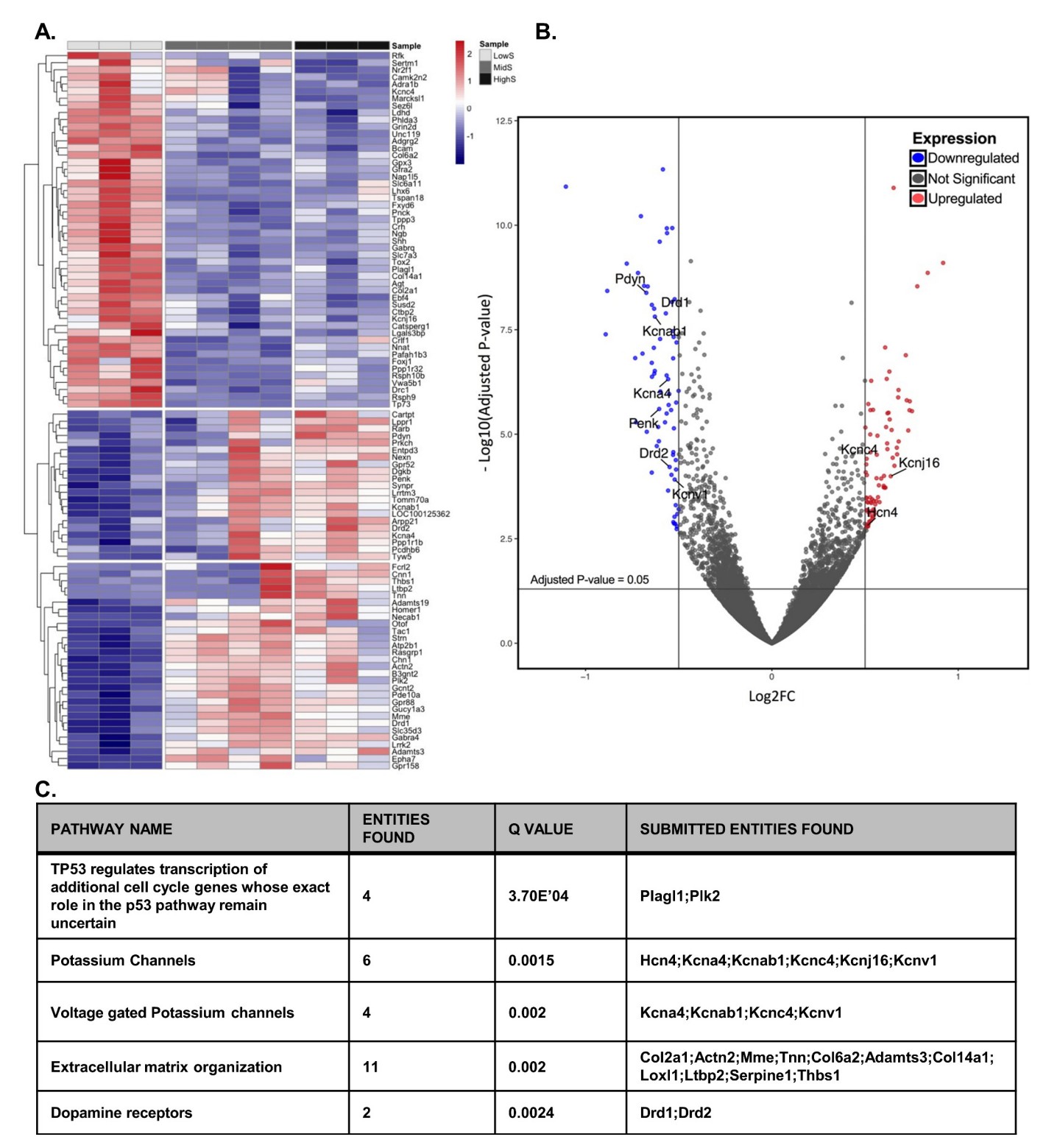

**Figure 3.** Divergent transcriptome profile in lowS and highS rats. (**A**) A clustergram summary of top differentially expressed genes (DEGs) between lowS, highS, and midS rats. Each column represents RNA sequencing of NAc tissue from a single animal. Log$_2$fold values are color coded red for upregulated genes and blue for downregulated genes. (**B**) Volcano plot highlighting genes related to K$^+$ channel activity and dopamine signaling in lowS versus highS transcriptome. (**C**) Pathway analysis showing top mechanistic networks related to divergent motivation for sucrose in lowS and highS rats (N = 3, 4 and 3 for lowS, midS and highS, respectively).

*Figure 3 continued on next page*

Figure 3 continued

DOI: https://doi.org/10.7554/eLife.47870.005

The following source data is available for figure 3:

**Source data 1.** Divergent transcriptome profile in highS and lowS rats.

DOI: https://doi.org/10.7554/eLife.47870.006

ANOVA) across a range of holding potentials indicating a similar number of channels underlying A-type currents and a similar profile of voltage-dependent activation of these channels. However, action potential output is strongly sensitive to $I_A$ inactivation kinetics (*Zemel et al., 2018*) with faster inactivation expected to increase action potential frequency similarly to the effect of reduction in the number of underlying channels (see Discussion). We found that the decay time of inactivation was 27% faster in lowS relative to highS animals (*Figure 4D*; $t_{(26)}$=2.1, p=0.047 t-test). We then looked at currents mediated by the large-conductance $Ca^{2+}$-activated potassium (BK) channels since they have been shown to interact with $I_A$ to influence both afterhyperpolarization and neuronal firing (*Kimm et al., 2015*; *Storm, 1987*; *Zhang and McBain, 1995*). Our slice recordings indicated no difference in BK current amplitude ($I_{BK}$) between lowS and highS groups (*Figure 4E*; [Main group effect: $F_{(1,12)}$=0.04, p=0.83], two-way RM ANOVA). There was also no difference in the total outward $K^+$ conductance measured after application of TEA and 4-AP (*Figure 4F*; Total $K^+$: [Main group effect: $F_{(1,13)}$=0.12, p=0.31], two-way RM ANOVA). Finally, recordings from midS animals aligned with findings in highS subjects: spike frequency was lower than the lowS group, but did not differ from highS animals, while $I_A$ amplitude, $I_{BK}$, and total $K^+$ currents did not differ between lowS, midS and highS groups (*Figure 4—figure supplement 2*). Notably, $I_A$ decay time in the midS group was intermediate to that of lowS and highS animals and not significantly different from either group (*Figure 4—figure supplement 2C*). Overall, our results suggest that a low motivation phenotype is linked to increased spiking of NAc principal neurons. This increased spiking is consistent with faster A-type current inactivation and decreased spike afterhyperpolarization, but is not related to availability of $I_A$ and BK channels or overall ion permeability through voltage-gated $K^+$ channels.

## $K_v$1.4 channels modulate SPN excitability selectively in lowS animals

Fast-inactivating A-type currents are mediated by $K^+$ channel proteins encoded by *Kcna4*, *Kcnc4*, or *Kcnd1-3* genes (*Coetzee et al., 1999*). The first two of these genes were represented in our sequencing results and code for $K_v$1.4 and $K_v$3.4 channels, respectively. Our $I_A$ recording protocol does not distinguish between these two $I_A$ channels. However, since only $K_v$1.4, but not $K_v$3.4, is strongly expressed in the NAc shell (*Pessia et al., 1996*; *Weiser et al., 1994*), we profiled $K_v$1.4 activity using an antagonist, UK-78,282. UK-78,282 exhibits ~200 fold selectivity at $K_v$1.4 over $K_v$3.4 and 10- to 700-fold selectivity at $K_v$1.4 over other closely related channels (*Kues and Wunder, 1992*). The overall effect of acute application of UK-78,282 (100 nM) was to suppress action potential output in lowS, but not highS, SPNs (*Figure 5A*). This differential effect reversed the excitability profile of lowS relative to highS neurons such that in the presence of UK-78,282, lowS SPNs fired fewer action potentials than highS SPNs (*Figure 5B*, [Main group effect: $F_{(1,12)}$=5.819, p=0.032] two-way RM ANOVA); cf. *Figure 4A*). UK-78,282 had a similar effect on afterhyperpolarization area in both lowS and highS animals (*Figure 5C*), suggesting that $K_v$1.4 activity does not account for differences in afterhyperpolarization between lowS and highS group at baseline (cf. *Figure 4B*). Additionally, UK-78,282 application did not result in significant differences between lowS and highS groups across a battery of excitability measures (*Figure 5—figure supplement 1*). Similar to highS animals, spike frequency in midS animals was not affected by UK-78,282 (*Figure 5—figure supplement 2A*). AHP in the midS group and $I_{BK}$ in either lowS, midS, or highS groups was also unaffected (*Figure 5—figure supplement 2B–E*).

We then evaluated contribution of $K_v$1.4 to A-type $K^+$ currents. Application of UK-78,282 significantly reduced amplitude of the $I_A$ in lowS, highS, and midS rats (lowS: [Main drug effect: $F_{(1,12)}$=14.03, p=0.0028]; highS: [Main drug effect: $F_{(1,13)}$=17.19, p=0.001] two-way RM ANOVAs; *Figure 5D*, *Figure 5—figure supplement 2F*). This was an unexpected result suggesting that reduced *Kcna4* transcript in lowS animals did not reduce availability of $K_v$1.4 at the cell surface relative to highS group. We did observe a difference in voltage-dependence of recorded currents

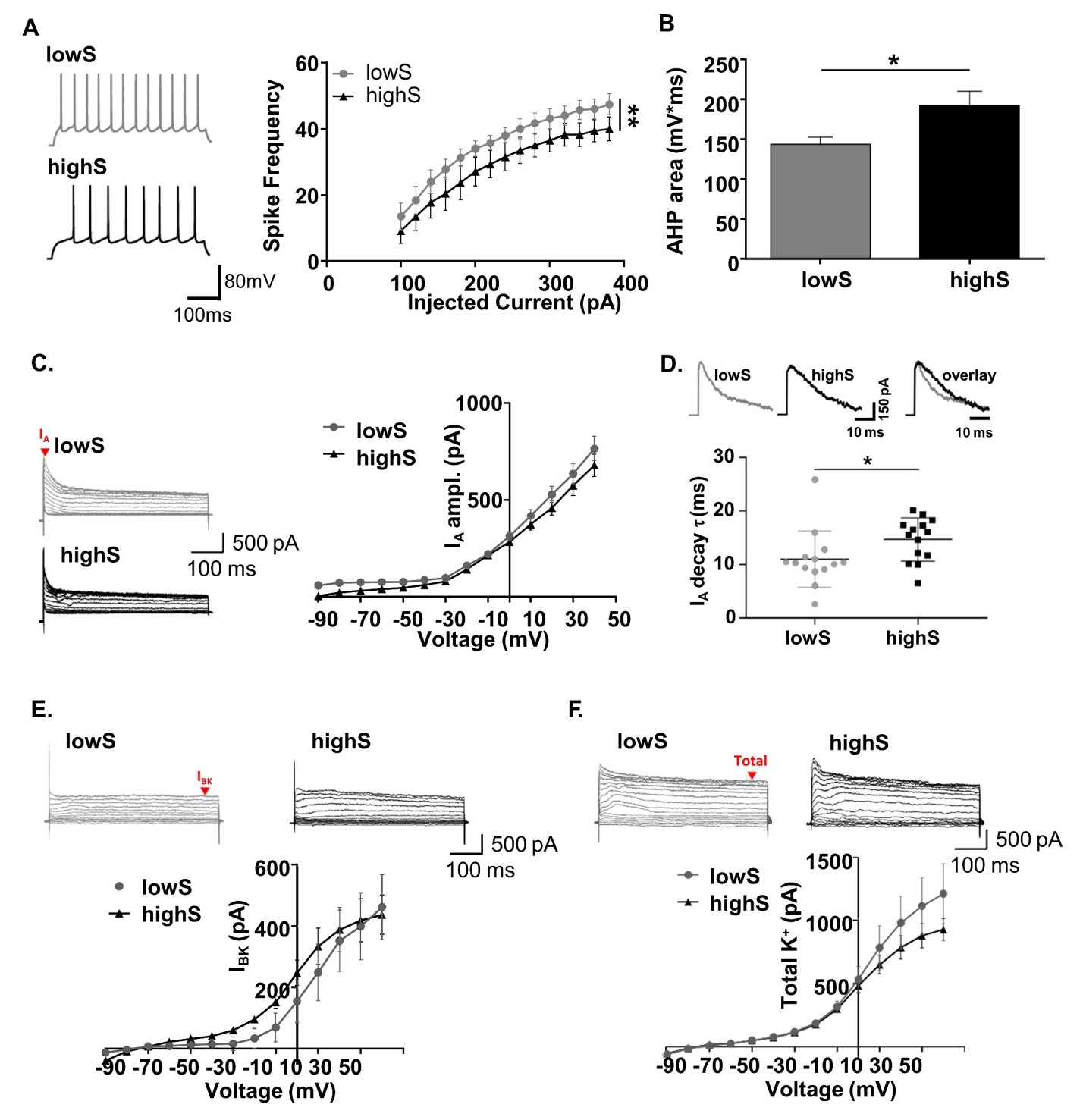

**Figure 4.** Low motivation for sucrose is associated with increased SPN excitability. (**A**) *Left*, representative traces from NAc SPNs in lowS and highS animals at a depolarizing (+200 mV) current step. *Right*, action potential output across a range of injected current steps is significantly elevated in the lowS group (**, p<0.01, n = 9 N = 5/group). (**B**) Decreased afterhyperpolarization area in SPNs from lowS animals is consistent with increased action potential firing (n = 9, N = 5/group; *, p<0.05). (**C**) *Left*, representative traces of A-type ($I_A$) currents in lowS and highS rats (n = 16 and 14 for lowS and highS, respectively; N = 5/group). Current amplitude was measured at the peak (red arrow). *Right*, Current-voltage relationship for $I_A$ is similar between groups. (**D**) *Top*, representative $I_A$ traces from lowS and highS animals are amplitude-scaled and overlaid to highlight differences in inactivation kinetics. *Bottom*, faster $I_A$ inactivation kinetics in the lowS animals in a scatterplot of decay times measured from the largest depolarizing peak in each group. (**E**) *Top*, representative traces of $I_{BK}$ currents isolated by paxilline. Current amplitude was measured at steady-state (red arrow). *Bottom*, there is no

*Figure 4 continued on next page*

*Figure 4 continued*

difference in $I_{BK}$ current-voltage relationship between lowS and highS animals (n = 7, N = 4/group). (F) *Top*, representative traces of total $K^+$ current blocked by combination of 4-AP and TEA. *Bottom*, there is no difference in current-voltage relationship for total $K^+$ current amplitude between lowS and highS animals (n = 7, N = 4/group).

DOI: https://doi.org/10.7554/eLife.47870.007

The following figure supplements are available for figure 4:

**Figure supplement 1.** Membrane excitability measures do not differ between lowS and highS groups.

DOI: https://doi.org/10.7554/eLife.47870.008

**Figure supplement 2.** Spike output and $K^+$ currents in midS animals align closely with the highS group.

DOI: https://doi.org/10.7554/eLife.47870.009

between groups: lowS neurons displayed sensitivity to UK-78,282 across all voltages, including the voltage range subthreshold to action potential firing, whereas $I_A$ in highS neurons was suppressed by UK-78,282 only at higher depolarizing voltages (*Figure 5D*). To follow-up on the observation that A-type currents display faster inactivation kinetics in lowS cells, we measured decay times of UK-78,282-sensitive currents. In lowS animals, application of UK-78,282 increased the time constant of inactivation from 7.32 ± 0.7 ms to 10.57 ± 1.5 ms (**Figure 5E**, $t_{(17)}$=3.5, p=0.003, paired t-test). In neurons from highS animals, $I_A$ inactivation was insensitive to UK-78,282 with the time constant of 9.07 ± 1.3 ms at baseline and 8.9 ± 1.3 ms after UK-78,282 application (**Figure 5E**, $t_{(13)}$=0.35, p=0.73, paired t-test). There was no effect of UK-78,282 on $I_A$ decay time in midS animals (*Figure 5— figure supplement 2G,$t_{(6)}$*=0.35, p=0.74, paired t-test). We conclude that differences in $K_v$1.4 gating, rather than number of available channels alters firing of NAc shell SPNs in a manner that may discriminate between behavioral extremes on a progressive ratio task.

## $K_v$1.4 blockade improves PR performance in lowS animals

Given the selective effect of UK-78,282 on neuronal excitability in lowS rats, we investigated whether a similar selectivity can be observed at the behavioral level. To examine this, we microinfused UK-78,282 into the NAc shell of lowS, midS and highS rats and measured its effect on PR performance. A two-way ANOVA with group (lowS, midS, highS) and UK-78,282 concentration (1 nM, 100 nM) as variables revealed a significant effect of group ($F_{(2,16)}$=153.6, p<0.0001), a significant effect of drug concentration ($F_{(1.97,31.5)}$ = 3.9, p=0.03) and significant interaction ($F_{(4,32)}$=5.1, p=0.003). To further explore these effects, we analyzed the effect of UK-78,282 infusion on each separate group. The low dose of UK-78,282 (1 nM) had no significant effect on sucrose self-administration in any group. A higher dose of UK-78,282 (100 nM), however, significantly increased breakpoints in the lowS, but not the highS or midS groups (lowS: $F_{(2,21)}$=9.5, p=0.001; midS: $F_{(2,25)}$=0.198, p=0.82; highS: $F_{(2,21)}$=0.5, p=0.64; one-way ANOVAs *Figure 6Ai,Bi;Ci*). Similarly, active lever responding and pellets earned were increased specifically in the lowS rats (*Figure 6Aii,iii, Bii,iii; Cii,iii*; active lever lowS: $F_{(2,21)}$=10.8, p=0.0006; active lever midS: $F_{(2,25)}$=0.1, p=0.9; active lever highS: $F_{(2,21)}$=0.05, p=0.95; pellets lowS: $F_{(2,21)}$=6.3, p=0.007; pellets highS: $F_{(2,25)}$=0.15, p=0.86; pellets highS: $F_{(2,21)}$=0.5, p=0.61; one-way ANOVAs). There was no difference in inactive lever presses between UK-78,282 concentrations in the lowS or midS groups (lowS: $F_{(2,21)}$=0.75, p=0.5; midS: $F_{(2,25)}$=0.14, p=0.86, one-way ANOVAs), although inactive lever pressing was significantly reduced by UK-78,282 (100 nM) in highS rats ($F_{(2,21)}$=6.1, p=0.008, one-way ANOVA; *Figure 6Aiv, Biv, Civ*). The reason for the latter finding is unclear, but it was strongly driven by a single rat robustly responding at baseline (15 inactive lever presses), but not after UK-78,282 treatment (one inactive lever press). Exclusion of this animal did not meaningfully affect highS breakpoint values, number of active lever presses, or pellets earned. Overall, these data show that an antagonist of $K_v$1.4 channels elevates willingness to exert effort for reward selectively in rats displaying lower motivation.

## Discussion

Our results show that a lower baseline motivation to expend effort for naturally reinforcing stimuli is linked to altered kinetics of a voltage-gated potassium channel, $K_v$1.4, in the NAc shell. Faster $K_v$1.4 inactivation accelerates action potential output of NAc SPNs, despite an apparent lack of changes in the number of functional $K_v$1.4 channels. Suppression of $K_v$1.4 activity by UK-78,282 selectively

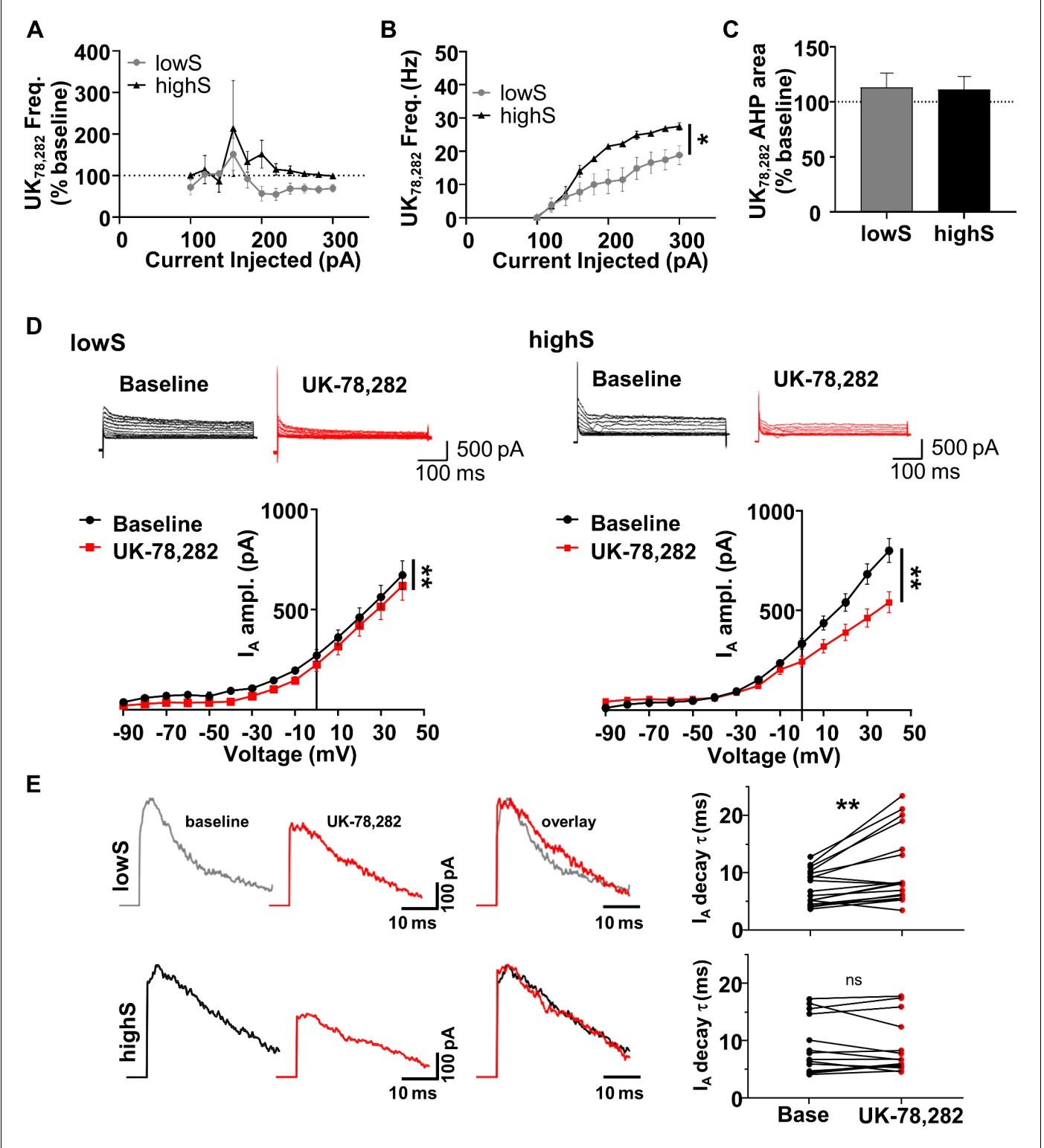

**Figure 5.** NAc neurons from lowS animals are uniquely sensitive to $K_v1.4$ antagonist, UK-78,282. (**A**) Action potential frequency was measured after application of UK-78,282 (100 nM) and expressed as percent change from frequency before UK-78,282 in the same cell (n = 7, N = 4/group). UK-72,282 suppresses firing in the lowS group to levels significantly different from baseline (Main drug effect: $F_{(1,6)}$ = 3.7, p=0.024, two-way RM ANOVA). In the highS group, firing after UK-72,282 application is not significantly different from baseline (Main drug effect: $F_{(1,6)}$ = 0.99, p=0.36, two-way RM ANOVA). (**B**) Firing frequency-current relationship highlights decreased action potential output in the lowS group after UK-78,282 application (*, p<0.05, two-way RM ANOVA). (**C**) UK-78,282 did not have an effect on AHP area in neurons from either the lowS or the highS groups. (**D**) *Top*, Representative traces of

*Figure 5 continued on next page*

*Figure 5 continued*

$I_A$ before and after UK-78,282 (100 nM) application in lowS (*left*, n = 13, N = 6) and highS (*right*, n = 14, N = 6) animals. *Bottom*, Current-voltage relationships indicate suppression of $I_A$ current amplitude by UK-78,282 in both groups (**, p<0.01, two-way RM ANOVA). (**E**) *Left*, representative $I_A$ traces before (baseline) and during UK-78,282 are amplitude-scaled and overlaid to highlight UK-78,282 effect on inactivation kinetics. *Right*, UK-78,282 increases $I_A$ inactivation time constant (τ) in lowS (top, n = 18, N = 6), but not highS (bottom, n = 16, N = 6), group. **, p<0.01, paired Student's t-test; ns, not significant. The highS data excludes two cells where UK-78,282 increased decay times to anomalous levels (cell 1: from 20.1 ms to 34 ms; cell 2: from 17.4 ms to 49.8 ms). Including these two cells in the analysis did not change statistical interpretation (p=0.23, paired Student's t-test).
DOI: https://doi.org/10.7554/eLife.47870.010
The following figure supplements are available for figure 5:

**Figure supplement 1.** UK-78,282 effects on membrane excitability measures.
DOI: https://doi.org/10.7554/eLife.47870.011
**Figure supplement 2.** UK-78,282 effects on spike output, AHP, and potassium channel currents.
DOI: https://doi.org/10.7554/eLife.47870.012

decreases SPN spiking and facilitates motivated behavior in subjects with a lower baseline motivation for reward.

## Selective effect of UK-78,282 on lowS $K_v$1.4

The lowS, but not the highS, phenotype that we describe was sensitive to the microinfusion of 100 nM concentration of UK-78,282 into the NAc. At this concentration, UK-78,282 is expected to be highly specific for $K_v$1.4. The target with the next nearest affinity, $K_v$1.3 channel, is blocked by UK-78,282 with an $IC_{50}$ of 280 nM (*Hanson et al., 1999*) and is not expressed in the striatum (*Kues and Wunder, 1992*). We were initially puzzled by the discordant electrophysiological findings with UK-78,282. The compound selectively suppressed action potential firing in the lowS group, but was equally efficacious at blocking the peak of A-type currents in both lowS and highS animals. Our observation that faster inactivation of A-type currents is also uniquely sensitive to UK-78,282 in the lowS group provided clues as to the potential mechanisms involved. For example, inactivation of $K_v$1.4 is regulated by phosphorylation via the calcium/calmodulin-dependent kinase II (CaMKII) that prolongs inactivation time course (*Roeper et al., 1997*). We detected no significant differences in expression of any of the four major CaMKII isoform genes (*Camk2a*, *Camk2b*, *Camk2d*, and *Camk2g*) in the RNA sequencing data. However, a number of reports indicate that D2 dopamine receptors stimulate CaMKII activity (*Takeuchi et al., 2002*; *Shioda and Fukunaga, 2017*). If this mechanism were to be involved, then decreased D2 receptor stimulation would lead to decreased CaMKII activity and faster A-type current inactivation. Further, activity of CaMKII is sensitive to the inhibitor proteins encoded by the *Camk2n1* and *Camk2n2* genes. Increased expression of these genes in the lowS animals would promote faster inactivation. Consistent with this, our sequencing data show significant downregulation of Drd2 and upregulation of *Camk2n2* (but not *Camk2n1*), transcripts in the lowS, relative to the highS, NAc tissue (*Figure 3A*, *Figure 3—source data 1*).

Differential assembly of the $K_v$1 channel tetramer may also play a role in regulating $K_v$1.4 signaling. For example, hetero-tetrameric assembly of $K_v$1.4 with the delayed rectifier subunits, $K_v$1.1 or $K_v$1.2, leads to slower inactivation relative to the $K_v$1.4 homomer (*Po et al., 1993*). $K_v$1.1 and $K_v$1.2, encoded by *Kcna1* and *Kcna2* genes, are both expressed in the rat striatum, albeit at lower levels than $K_v$1.4 (*62*). Neither *Kcna1* nor *Kcna2* were differentially expressed in the lowS vs highS sequencing data. However, expression of the regulatory subunit $K_v$β1 (*Kcnab1* gene) that promotes cell surface expression of $K_v$1.x heterotetrameric complexes (*Manganas and Trimmer, 2000*) was markedly lower in the NAc of lowS animals. Lower expression of $K_v$β1 is expected to decrease availability of $K_v$1.x hetero-tetramers and contribute to faster $K_v$1.4 channel inactivation reported here.

Both behavioral and electrophysiological effects of UK-78,282 in midS animals were aligned closely with effects observed in the highS group. It is intriguing that with regard to possible regulators of $K_v$1.4 function discussed above, RNA sequencing data from midS animals indicated significantly lower levels of *Kcna1* and *Kcnab2* transcripts expected to lead to faster inactivation, but decreased *Camk2n1* (no change in *Camk2n2*) levels expected to lead to slower inactivation (*Figure 3—source data 1*). Electrophysiological data indicated that $I_A$ decay time in midS animals at baseline was intermediate to that of lowS and highS groups (*Figure 4—figure supplement 2C*), however midS animals were insensitive to UK-78,282 at both the $I_A$ kinetics and behavioral levels.

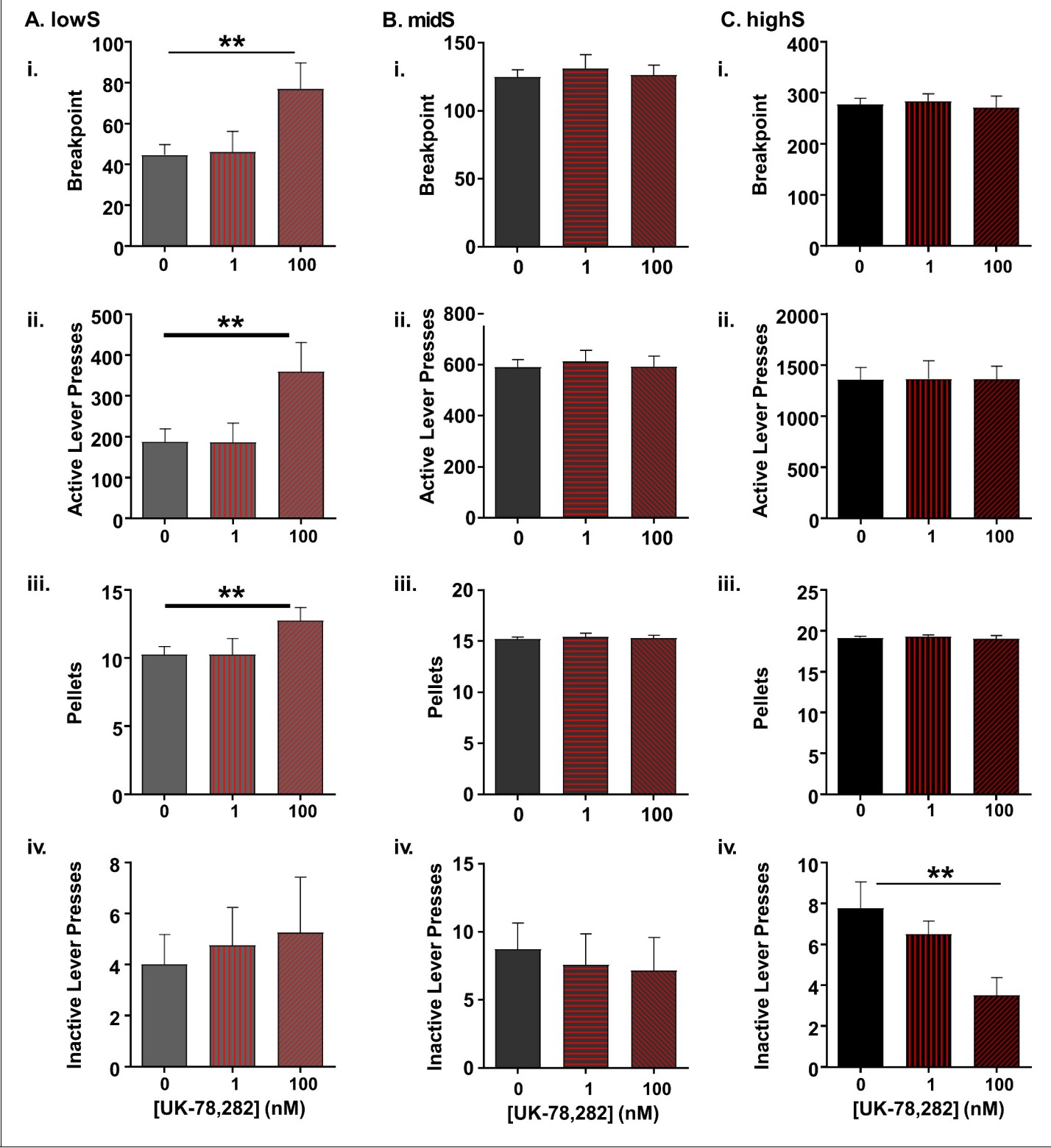

**Figure 6.** Selective effect of K$_v$1.4 antagonism on PR performance in lowS animals. (**A**) In lowS animals (N = 6), microinjection of UK-78,282 into the NAc shell dose-dependently increased: i) breakpoints, ii) active lever presses and iii) pellets earned, but not iv) inactive lever presses. (**B**) In midS animals (N = 7) neither 1 nM nor 100 nM concentration of UK-78,282 had an effect on i) breakpoints, ii) active lever presses, iii) pellets earned or iv) inactive lever presses following 100 nM UK-78,282 microinjection. (**C**) In highS animals (N = 6) neither 1 nM nor 100 nM concentration of UK-78,282 had an effect on i)

*Figure 6 continued on next page*

*Figure 6 continued*

breakpoints, ii) active lever presses, or iii) pellets earned. There was a significant reduction in iv) inactive lever presses following 100 nM UK-78,282 microinjection. **, p<0.01, one-way ANOVAs.

DOI: https://doi.org/10.7554/eLife.47870.013

The following figure supplement is available for figure 6:

**Figure supplement 1.** Histological verification of placements for UK-78,282 microinjection experiments.

DOI: https://doi.org/10.7554/eLife.47870.014

These findings highlight remarkable flexibility and multiple redundancies that can impact activity of a single channel with possible behavioral consequences. Other possibilities for regulation of $K_v1.4$ inactivation include interactions with intracellular heme, intracellular pH, protein phosphatases, and membrane lipids (*Roeper et al., 1997*; *Sahoo et al., 2013*; *Padanilam et al., 2002*; *Oliver et al., 2004*). Direct examination of each of these mechanisms is outside the scope of this work. We can conclude, however, that unique sensitivity of lowS NAc neurons to UK-78,282 is likely conferred by differential interaction of $K_v1.4$ channels with binding partners or phosphorylation mechanisms rather than functional availability of $K_v1.4$ channels on the cell surface.

## $K_v1.4$ regulation of neuron firing

Application of UK-78,282 suppressed peak $I_A$ amplitude in both lowS and highS animals (*Figure 5D*). However, there was a difference in voltage-dependence of the enhancement. In the lowS animals, our results indicate equal availability of $K_v1.4$ across potentials, including in the zone subthreshold to action potential firing. In the highS animals, $K_v1.4$ availability is greater at suprathreshold potentials. It is not clear what contributes to this difference. The voltage-dependence of activation kinetics of $K_v1$ family members has been shown to shift in the depolarizing direction by assembly with $K_v1.2$ subunit (*Baronas et al., 2015*). Moreover, $K_v1.2$ has been shown to enhance activation of $K_v1.4/1.2$ heterotetramers following depolarizing pre-pulses, a phenomenon termed use-dependent activation (*Baronas et al., 2015*). Such use-dependence may play a role in suppressing action potentials during spike trains and contribute to lower action potential spike frequency observed in highS animals. Slower inactivation of $I_A$ in highS animals, discussed above, will also tend to suppress action potential output.

The effect of constitutive $K_v1.4$ deletion on action potentials has been examined in two previous studies. The first one of these found that cortical pyramidal neurons of $K_v1.4^{-/-}$ mice have shorter action potential width, but similar resting membrane potential, input resistance and rheobase relative to wild-type controls (*Carrasquillo et al., 2012*). Blockade of $K_v4$ channels in this study, however, unmasked differences across a broader range of membrane properties, indicating involvement of $K_v4$-mediated compensatory mechanisms. The second report, found increased firing of suprachiasmatic nucleus neurons using the same line of knock-out mice, but did not evaluate other measures of membrane excitability (*Granados-Fuentes et al., 2012*). In a study of NAc neurons, suppression of A-type currents by dopamine was also linked to increased firing, although contribution of $K_v1.4$ to these currents was not specifically examined (*Hopf et al., 2003*). Increased action potential frequency in $K_v1.4^{-/-}$ mice is consistent with our recordings from SPNs of the NAc in lowS animals, given the RNA sequencing data that indicates decreased levels of *Kcna4* transcript in this group. However, we find no differences in action potential width between lowS (1 ± 0.04 ms) and highS (0.99 ± 0.07 ms) groups. Indeed, we observe that lowS and highS groups are similar across a broad spectrum of action potential and intrinsic excitability measures both in the absence and in the presence of UK-78,282 (*Figure 4—figure supplement 1*, *Figure 5—figure supplement 1*). Taken together, these observations argue for brain-region specific impact of $K_v1.4$ on neuronal output that is additionally guided by interactions with other channels or modulatory mechanisms.

## Spike frequency and motivation for reward

The relationship between NAc spiking and reward seeking has been a subject of intense interest for decades. There is general support from behavioral pharmacology studies that inhibition of NAc shell promotes seeking of natural reward (*Basso and Kelley, 1999*; *Reynolds and Berridge, 2001*; *Kelley and Swanson, 1997*). There are also strong indications that NAc firing may be modulated by

reward-associated cues. For example, reward consumption has been shown to inhibit firing of NAc shell neurons in vivo, but only in the presence of cues predicting reward, whereas sustained increase in NAc firing has been proposed to inhibit operant responding for natural reward (*Nicola et al., 2004a*). Multiple other behavioral cues regulate NAc firing in vivo, including timing of reward delivery, magnitude of reward, and reward identity (*Nicola et al., 2004a*; *Nicola et al., 2004b*; *Taha et al., 2007*; *Krause et al., 2010*). Our data are unique in that we report SPNs from animals less motivated to pursue reward to be biased toward higher firing in the slice, in the absence of external behavioral cues.

An obvious suspect for differences in spike output in our lowS and highS datasets is relative abundance of D1-expresing and D2-expressing neurons. D1- and D2-expressing SPNs display distinct electrophysiological properties with D2 neurons showing greater excitability and lower threshold for action potential firing than D1 neurons (*Cepeda et al., 2008*; *Gertler et al., 2008*; *Ade et al., 2008*). Meanwhile, behavioral data provides mixed clues with some evidence supporting D1 SPNs as mediators of positive aspects of reward, and D2 SPNs as mediators of behavioral aversion (*Kravitz et al., 2012*; *Hikida et al., 2010*), while others report that signaling at both D1 and D2 SPNs enhances motivation for natural rewards (*Soares-Cunha et al., 2016*). Relative contribution of D1- and D2- SPNs to our data is not known and we have not directly addressed this question. However, several considerations argue against biased sampling, including random selection of SPNs from the pool of visually identified neurons and similar input resistance and rheobase values, hallmarks of D1 or D2 identity (*Gertler et al., 2008*; *Janssen et al., 2009*), between groups. RNA sequencing results indicated reduced expression of both D1 dopamine receptor and D2 dopamine receptor encoding genes in lowS relative to highS animals, rather than a selective reduction of one receptor population. If maintained at the protein level, this reduction highlights additional possibilities for regulation of neuronal output as a function of effort-based motivation.

Effort-based motivation for natural reward is unlikely to depend on a uniformly sustained increase or decrease in firing across all NAc shell SPNs. During a progressive ratio task for a cocaine reinforcer, a transient increase in NAc firing in vivo has been proposed to serve as a behavioral breakpoint signal, potentially driven by cocaine-induced increases in dopamine levels (*Nicola and Deadwyler, 2000*). However, behavioral output during progressive ratio for cocaine involved some neurons that are excited and others that are inhibited throughout the different phases of a behavioral task (*Nicola and Deadwyler, 2000*). Indeed, responses combining both excitation and inhibition have been observed in the NAc across many reward-seeking behaviors examined with in vivo electrophysiology. It is possible that such dynamic responses are guided by interactions with a distinct pattern of inputs to or outputs from the NAc and our experiments do not address this contingency. Our data do support the idea that any behavior encoded by NAc shell output will be biased by greater intrinsic likelihood of generating such output by SPNs in lowS, relative to highS, animals.

## Conclusions

In summary, we describe a set of neuronal and genetic features associated with motivation to exert effort for natural reward. Lower motivation is linked to a divergent transcriptome profile, and increased SPN output in the NAc shell. Increased SPN output depends on faster inactivation kinetics of $K_v1.4$ and blockade of $K_v1.4$ activity selectively increases effort for natural reward in animals displaying low motivation. These results point to modulators of $K_v1.4$ gating as potential targets in a broad spectrum of psychiatric disorders associated with deficits in motivation.

## Materials and methods

### Subjects

Male Sprague-Dawley rats (Rattus norvegicus), weighing between 250–300 grams (Taconic Laboratories, Germantown, NY, USA) were individually housed in a colony room, rats were food restricted (20 g normal chow per day) with *ad libitum* water access and were maintained on a 12 hr/12 hr light/dark cycle, with lights on at 0700 hr. All experimental procedures were followed in accordance with the University of South Carolina School of Medicine and University of Kentucky Institutional Animal Care and Use Committees.

## Sucrose self-administration

Self-administration experiments were conducted in ventilated, sound attenuating operant chambers, equipped with a house light, active and inactive response levers, a pellet dispenser and a food receptacle (Med-Associates Inc, East Fairfield, VT, USA). During the experiment, rats had *ad libitum* access to water and were fed 20 g of normal chow per day after the operant session. All subjects were trained daily on a fixed-ratio one (FR1) schedule of reinforcement, in which each active lever press delivered a single 45 mg sucrose pellet. Presses on the inactive lever had no programmed consequences. Each sucrose pellet was followed by a 20 s timeout period during which house light went off and lever responses had no scheduled consequences. Once stable responding was achieved under the FR1 schedule, the rats progressed to the FR3 schedule (three active lever presses for one sucrose pellet) and then to FR10 (ten active lever presses for one sucrose pellet). Once stable under the FR10 schedule, subjects were placed on a progressive ratio (PR) schedule of reinforcement during which successive reinforcements could be earned according to an increasing number of lever-presses based on the formula: $[5e^{(pellet \# * 0.2)}] - 5$ (*Richardson and Roberts, 1996*). The session ended when rats failed to reach the next lever-press criterion within 1 hr. The final ratio achieved was recorded as the 'breakpoint' value. Rats were run on the PR schedule until stable responding was achieved. Stable responding under both FR and PR schedules was defined as <10% variability in active lever responses over three consecutive daily sessions.

## Tissue collection

Tissues were harvested from animals 24 hr after the final behavioral session. Trunk blood and tissue punches from the NAc shell region were collected. Punches were flash frozen on dry ice and plasma was separated from trunk blood after centrifugation (3200 rpm) at 4°C. Tissue and plasma were stored at −80°C.

## Plasma corticosterone analysis

Plasma corticosterone was measured using a corticosterone ELISA kit (Enzo Life Sciences, Farmingdale, NY). Plasma was diluted 1:40 and run according to manufacturer's protocol. Plates were read using a Synergy 2 Multi-Mode plate reader (Bio Tek, Winooski, VT) with Gen5 software (Bio Tek, Winooski, VT).

## RNA sequencing

Total RNA was extracted using TRIzol (Life Technologies) and a RNeasy Mini Kit (Qiagen) according to manufacturer's instructions. Samples were homogenized and incubated in TRIzol for 5 min before addition of chloroform and vigorous shaking for 30 s. Following a 3 min incubation, samples were centrifuged at 4°C for 10 min at maximum speed ($\geq$10,000 rpm). The aqueous phase was aspirated and transferred to a microcentrifuge tube before addition of 70% EtOH, centrifugation at $\geq$10,000 rpm for 15 s, and collection of the precipitate from the RNeasy mini column. This step was repeated after adding 700 µL Buffer RW1 and, next, 500 µL Buffer RPE to the mini column. Another 500 µL of Buffer RPE was centrifuged for 2 min before the sample/mini column underwent a 2 min 'dry' spin and transferred to the final collection tube. Last, 30 µL DEPC water was used to elute the sample. RNA samples were quantified using NanoDrop Spectrophotometer ND-2000 (Nanodrop Technologies) and checked for quality and degradation by Agilent 2100 Bioanalyzer. All samples were of high quality (RNA integrity numbers between 9.9 and 10). Strand-specific mRNA libraries were prepared using the TruSeq Stranded mRNA Library Prep Kit (Set B, Illumina Inc) and sequenced on the Illumina NextSeq500 in a paired-end mode with read length of $2 \times 75$ bp.

## Sequencing data preprocessing and analyses

To ensure there were no sequencing errors, raw sequences were checked for quality using FastQC, and then aligned to the rat genome (downloaded from iGenomes, Illumina) using the STAR aligner program (*Dobin et al., 2013*). Aligned SAM files from STAR were converted to BAM files using SAMtools (*Li et al., 2009*). BAM files were processed for read summarization using featureCounts (*Liao et al., 2014*), and the resulting read counts were preprocessed by filtering out low read counts (read counts < 5) in R software. Processed data were then analyzed for differential expression using DESeq2 (*Love et al., 2014*) in R software. False discovery rate (FDR < 0.05) was used to determine

the threshold of p- value for the analysis. Functional annotation/gene ontology analyses for biological function were conducted using the Reactome classification system (https://reactome.org/) accessed in February-March, 2018. Reactome is an open-source, curated database of biological pathways and processes (*Fabregat et al., 2018*; *Milacic et al., 2012*).

## Electrophysiology

Rats were decapitated following isoflurane anesthesia 24 hr after the last behavioral session. Brains were rapidly removed and coronal slices (300 μm-thick) containing the NAc were cut using a Vibratome (VT1200S; Leica Microsystems) in an ice-cold aCSF solution in which NaCl was replaced with an equiosmolar concentration of sucrose. ACSF contained the following (in mM): 130 NaCl, 3 KCl, 1.25 $NaH_2PO_4$, 26 $NaHCO_3$, 10 glucose, 1 $MgCl_2$, and 2 $CaCl_2$; pH 7.2–7.4, when saturated with 95% $O_2$ and 5% $CO_2$. Slices were incubated in aCSF at 32–34°C for 45 min and kept at 22–25°C thereafter, until transfer to the recording chamber. All solutions had osmolarity between 305 and 315 mOsm. Slices were viewed under an upright microscope (Olympus BX51WI) with infrared differential interference contrast optics and a 40 × water immersion objective. For recordings, the chamber was continuously perfused at a rate of 1–2 ml/min with oxygenated aCSF heated to 32 ± 1°C using an automated temperature controller (Warner Instruments). Recording pipettes were pulled from borosilicate glass capillaries (World Precision Instruments) to a resistance of 4–7 MΩ when filled with the intracellular solution. The intracellular solution contained the following (in mM): 145 potassium gluconate, 2 $MgCl_2$, 2.5 KCl, 2.5 NaCl, 0.1 BAPTA, 10 HEPES, 2 Mg-ATP, and 0.5 GTP-Tris; pH 7.2–7.3, with KOH; osmolarity 280–290 mOsm.

NAc shell SPNs were identified by their morphology and low resting membrane potential (RMP, −70 to −85 mV) and voltage-clamped at −70 mV. Current step protocols (from −500 to +500 pA; 20 pA increments; 500 ms step duration) were run to determine action potential frequency versus current (*f-I*) relationships. $K^+$ currents were recorded in voltage-clamp mode with 1 mM QX-314 added to the intracellular solution. Following seal rupture, series resistance was compensated (65–75%). Outward currents were evoked by incrementing holding voltage from −90 mV to +40 mV in 10 mV steps. This protocol was then repeated with a 100 ms pre-step to a depolarized potential (−40 mV) at which $I_A$ currents are inactivated. Currents recorded after the −40 mV pre-step were subtracted from those recorded without the pre-step in the same cell, yielding $I_A$ that was measured at the peak of subtracted current. BK currents were measured at steady-state after subtracting membrane current responses in the presence of BK channel antagonist, paxilline (10 μM), from responses recorded in the absence of paxilline in the same cell. Total $K^+$ currents were defined as those sensitive to combined application of 4-AP (0.5 mM) and TEA (10 mM) as previously described (*Ji and Martin, 2014*) and measured at steady-state. Drugs were applied via the Y-tube perfusion system modified for optimal solution exchange in brain slices (*Hevers and Lüddens, 2002*). All data were collected after a minimum of 2 min of drug exposure. Currents were low-pass filtered at 2 kHz and digitized at 20 kHz using a Digidata 1440A acquisition board (Molecular Devices) and pClamp10 software (Molecular Devices). Access resistance (10–30 MΩ) was monitored during recordings by injection of 10 mV hyperpolarizing pulses; data were discarded if access resistance changed >25% over the course of data collection. All analyses were completed using Clampfit 10 (Molecular Devices).

## In vivo microinjections

A subset of rats was implanted with guide cannulas targeting the NAc shell prior to training on a PR schedule of reinforcement. Following isoflurane anesthesia (2–5% isoflurane in $O_2$) animals were placed in a stereotaxic instrument (Kopf Instruments, Tujunga, CA, USA), and guide cannulas targeting the NAc shell were positioned using the following stereotaxic coordinates (in mm from bregma): +1 AP, ± 1 ML, −5 DV. Guide cannulas were fixed to the skull with dental acrylic and stainless-steel obturators were placed inside the cannulas to prevent occlusions. Following a 7 day recovery period, animals proceeded to PR training and once stable PR responding was achieved, they underwent two microinjection sessions. During the first session, animals received a bilateral infusion of either UK-78,282 (1 nM) or UK-78,282 (100 nM) into the NAc shell through microinjectors extending 2 mm below tips of the guide cannulas. Microinfusions were at 0.5 μl/side over 2 min plus one minute of passive diffusion away from cannula tips. The criterion performance on the PR schedule was then re-

established over consecutive daily sessions. After that, the animals underwent a second test session during which they received microinjection of a different UK-78,282 concentration. Microinjections of the two UK-78,282 concentrations were counterbalanced between animals and no animal received more than two microinjections. Cannula placements were confirmed histologically by cresyl violet staining (*Figure 6—figure supplement 1*).

## Statistics

Statistical analyses were performed with Excel 2016 (Microsoft) or GraphPad Prism 6 (GraphPad software). For behavioral and electrophysiological experiments, Students t-tests, one-way ANOVAs, or two-way repeated measures ANOVAs followed by Bonferroni post hoc tests were performed as indicated in the text. Sample sizes were determined using G power 3.1.9.4 (effect size = 0.5, alpha = 0.05, power = 0.8). Throughout the manuscript, cell numbers are designated 'n', while animal numbers are designated 'N'. Data were reported as mean ± standard error of the mean and statistical significance thresholds were set at $p < 0.05$.

## Acknowledgements

Funding Support: This work was supported by the National Institutes of Health grants K01DA031747, R01DA041513 (PIO), R00DA032681, R01DA044311 (JRT), T32DA016176 (RDC).

## Additional information

### Funding

| Funder | Grant reference number | Author |
| --- | --- | --- |
| National Institute on Drug Abuse | DA031747 | Pavel I Ortinski |
| National Institute on Drug Abuse | DA041513 | Pavel I Ortinski |
| National Institute on Drug Abuse | DA032681 | Jill R Turner |
| National Institute on Drug Abuse | DA044311 | Jill R Turner |
| National Institute on Drug Abuse | DA016176 | Robert D Cole |

The funders had no role in study design, data collection and interpretation, or the decision to submit the work for publication.

### Author contributions

Bernadette O'Donovan, Conceptualization, Data curation, Software, Formal analysis, Validation, Investigation, Visualization, Methodology, Writing—original draft, Project administration, Writing—review and editing; Adewale Adeluyi, Data curation, Software, Formal analysis, Visualization, Methodology; Erin L Anderson, Formal analysis, Methodology; Robert D Cole, Data curation, Methodology, was added to the manuscript at the review stage, and provided essential contributions to acquisition and curation of the new data requested by the reviewers of the original submission; Jill R Turner, Resources, Formal analysis, Supervision, Funding acquisition, Investigation, Visualization, Methodology, Writing—original draft, Writing—review and editing; Pavel I Ortinski, Conceptualization, Resources, Data curation, Software, Formal analysis, Supervision, Funding acquisition, Validation, Investigation, Visualization, Writing—original draft, Project administration, Writing—review and editing

## Author ORCIDs

Bernadette O'Donovan (iD) https://orcid.org/0000-0002-3591-0560
Erin L Anderson (iD) https://orcid.org/0000-0002-4731-001X
Pavel I Ortinski (iD) https://orcid.org/0000-0003-0814-4490

## Ethics

Animal experimentation: All of the animals were handled according to approved institutional animal care and use committee (IACUC) protocols #2324-101167-120116 of the University of South Carolina and #2018-3132 of the University of Kentucky.

## Decision letter and Author response

Decision letter https://doi.org/10.7554/eLife.47870.017
Author response https://doi.org/10.7554/eLife.47870.018

## Additional files

### Supplementary files

• Transparent reporting form
DOI: https://doi.org/10.7554/eLife.47870.015

### Data availability

Data generated or analysed during this study are included in the manuscript and supporting files.

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
