## [Decision Letter]

Thank you for submitting your article "Altered gating of K_v_1.4 in the nucleus accumbens suppresses motivation for reward" for consideration by *eLife*. Your article has been reviewed by three peer reviewers, one of whom is a member of our Board of Reviewing Editors, and the evaluation has been overseen by Ronald Calabrese as the Senior Editor. The following individuals involved in review of your submission have agreed to reveal their identity: Kauê Machado Costa (Reviewer #2); Alexxai Kravitz (Reviewer #3).

The reviewers have discussed the reviews with one another and the Reviewing Editor has drafted this decision to help you prepare a revised submission.

Summary:

The reviewers found that the study used a strong, multifaceted approach to reveal genetic and physiological underpinnings of population variance in effort-based motivation. The main findings are:

1) There is substantial variability in effort-based motivation.

2) Those with high v. low effort-based motivation have a different transcriptional profile in the NAc shell, including differential expression of 4 genes related to dopamine signaling and 6 related to K^+^ channels.

3) Those with low v. high effort-based motivation show lower electrically-evoked phasic dopamine release in the NAc shell.

4) NAc shell medium spiny neurons from low motivation rats were more excitable, with smaller AHPs and faster IA inactivation. This was linked to changes in K_v_1.4, a transcript downregulated in low motivation rats.

5) Inactivation of K_v_1.4 in the NAc shell increased motivation selectively in low motivation rats.

Essential revisions:

1) The sucrose preference test data are underpowered and, moreover, these data do not reflect the hedonic experience of the pellets earned during the progressive ratio task. For that reason, the reviewers suggested removing these data. All reviewers agreed that these data were not critical, and were even tangential to the main point of this study. If they are to be kept in, power should be added and their interpretation shifted to avoid the implication that these data speak to the hedonic experience of the reward in the PR task.

2) The experiments assessing the effects of UK-78,282 on IA inactivation (Figure 6E) are also underpowered. Because this is such a vital component of the study, the reviewers would like to see the N for both highS and lowS subjects increased.

3) Reviewers were also concerned with the power in the FSCV and FSCAV experiments. For the FSCAV data this was especially a problem with the marginal and thus unclear effect. They agreed that these experiments are also not critical to the conclusions drawn here and can be removed. If you choose to keep them in, power should be added so both the evoked FSCV and tonic FSCAV results are more convincing.

4) The reviewers agreed that is not possible to adequately differentiate D1 and D2 MSNs by electrophysiological criteria alone. Thus, the claims in the Discussion, which were not supported with data, of no correlation between putative D1 or D2 identity and electrophysiological responses should be removed and instead the Discussion should make clear to the reader that this remains an open question.

5) Because motivation in the PR task is interpreted as a stable trait-like feature, the reviewers agreed that it would be helpful provide data on the stability of PR performance across days.

6) Please clarify how lowS, midS, and highS groups were defined for individual experiments and report the behavioral data for each smaller cohort.

Reviewer #1:

The data in this report demonstrate substantial variability in motivation to exert effort in a progressive ratio task for food reward. Taking the top and bottom quartile of the subjects, the authors found that those subjects with low PR performance showed a different NAc shell transcriptional profile than those with high performance, had lower phasic dopamine released evoked by electrical stimulation, and higher MSN excitability. The latter was linked to K_v_1.4 function and K_v_1.4 blockade was found to increase motivation in low PR subjects. The multifaceted approach here is a strength. There are a lot of data here that will be of interest to many researchers to follow up on. The findings are novel and exciting, demonstrating a new potential mechanism of effort-related motivation and therapeutic target. My concerns are noted below.

1) There is substantial variability in PR performance, leading to categorization of low, and high S groups for subsequent experiments, which seems reasonable. But it is not clear within these smaller cohorts used for the RNA-seq, FSCV, physiology, and infusion experiments how low, mid, and high were defined. Was this top and bottom 25% within each smaller cohort, or within the larger group? This should be made clear and the behavioral data for each smaller cohort should be shown, perhaps in a supplementary figure.

2) Because motivation in the PR task is interpreted as a stable trait-like feature, it would be helpful to show that it was consistent within subject across days.

3) To say that K_v_1.4 blockade 'rescues' a deficit in lowS animals is a bit of an overstatement, because even with the blockade breakpoint, presses, and earned pellets remain lower in the lowS than the midS group. To make this clear, the midS data should be included in the main Figure 7. Moreover, to make an interpretation that K_v_1.4 blockade had an effect in one group but not another, the data should be analyzed together in a 2 way ANOVA with variables group (low, mid, high S) and drug dose. A significant interaction would justify the individual group ANOVAs evaluating dose within each group.

4) The FSCV methods are not adequately described. The frequency of MFB electrical stimulation needs to be provided. There should be some description of how electrode placements were optimized. The FSCAV methods are not provided. Details on how the FSCV data were analyzed, including how the max was determined, how the half-life was calculated, and how transient events were defined and calculated, need to be provided.

5) The 'hedonic preference' findings lead to the, perhaps unintended, implication that PR performance and reward hedonic experience are not related. But sucrose preference in this two bottle test is not reflective of the hedonic preference of the pellets earned in the PR task and so this interpretation could be somewhat misleading. I suggest referring to this as sucrose preference, as is more common, and walking back conclusions that PR and hedonia were unrelated.

6) The N for the FSCV data are somewhat low in the lowS group (N=5), which becomes a problem for the marginal tonic DA effect. I'm concerned the data are not sufficiently powered to detect an effect. If these data are going to be shown, power should be added or a power analysis provided to indicate these experiments were sufficiently powered.

Reviewer #2:

In their study, O'Donovan et al. set out to identify potential genetic and physiological underpinnings of population variance in effort-based motivation. They use PR breakpoints as an index for this behavioral variable in a naïve cohort of Sprague-Dawley rats, focusing on differences between animals at the top (highS) and bottom (lowS) quartiles of the PR breakpoint distribution. First, they showed that these groups were similar in their FR responses, sucrose preference (but see comment below) and stress hormone levels. Authors performed a genome-wide RNA sequencing of tissue samples from the nucleus accumbens (NAc) of these groups as a broad-scope strategy for identifying differentially expressed transcripts. Out of the 164 identified genes differentially expressed in low- versus high-motivation rats, the authors found four genes related to dopamine signaling and six related to K^+^ channels. Using in vivo FSCV, they show that lowS rats have lower evoked dopamine transients in the NAc shell. Furthermore, striatal projection neurons (SPNs) from lowS rats were more excitable, with smaller AHPs and faster IA inactivation. They pharmacologically demonstrate that this phenotype is linked to changes in K_v_1.4 (KCNA4), which was identified as a downregulated transcript in their RNA screening. The changes in K_v_1.4 currents seen in lowS versus highS animals were not simply related to current amplitude, as would perhaps have been expected based on the RNA screening alone, but involved a specific acceleration of IA inactivation that was selectively impacted by blockade with UK-78,282. The authors speculate on the potential mechanism underlying this finding but chose not to explore it within this study. Finally, they pharmacologically inactivate K_v_1.4 within the NAc shell in vivo, resulting in a dose-dependent increase in motivation selectively for the lowS rats, establishing a causal link between gene expression variability, K_v_1.4 channels, SPN excitability in the NAc shell and behavioral motivation.

Overall, the study is very interesting, well conducted and of high relevance. The approach used to isolate potential underpinnings for behavioral variability is at the forefront of the field. The core of the presented data is quite adequate, and the findings are truly exciting. However, a few key issues need to be addressed.

One major concern is that the sucrose preference test is woefully underpowered. It seems as if only 3 highS and 3 or 4 lowS rats were included in this test, and there is a clear trend towards the lowS animals having a higher sucrose preference. This is particularly important given the role of specific subregions of the NAc shell in mediating hedonic preference vs. incentive salience (as established in the work of Kent Berridge and others). The authors unfortunately would need to repeat this test, specifically with more highS and lowS rats, in order to have a conclusive result and rule out potential differences in sucrose preference.

The authors also claim to have clustered their recorded cells into putative D1- and D2-expressing SPNs based on their electrophysiological properties and saw no difference in "electrophysiological responses at baseline, or following drug application" between the groups, but this data is not shown. This analysis and its results must be disclosed, as well as the relative number of putative D1 and D2 cells in each experiment, wherever this is possible. Any differential effect between these two cells types, as well as potential sampling biases, could lead to major changes in the interpretation of the results. Results reported in Figure 6E are of particular concern, given that the selective effect of UK-78,282 on IA inactivation in the lowS group (a key finding of the study) is "demonstrated" with less than 6 cells in each group, which is already a relatively underpowered experiment without considering a potential asymmetry in the sampling of D1 vs. D2 subpopulations.

Reviewer #3:

The manuscript by O'Donovan and colleagues addresses an important question of individual variability among rats on motivational tasks, specifically here in the progressive ratio task. Variance is often high on these tasks, and reasons for this inter-animal variance are not known. In an elegant set of studies, the authors link this variance to changes in the transcriptional landscape of the nucleus accumbens, and drill down further to link it to changes in potassium channel expression. They highlight one channel in particular, K_v_1.4, and perform extensive electrophysiology to show that it is differentially active in low vs. high motivated rats. This channel is expressed highly in both dorsal and ventral striatum, and the authors further demonstrate that pharmacological manipulations of its function can increase motivation in rats that are on the low end of PR responding. Overall, I found the study well designed and experimentally sound, and the discussion of the questions and implications were both interesting and appropriate. I have only suggestions for improvement.

1) In Figure 1, they show that the differences in PR were not observable in the preceding FR sessions. They argue this by looking at average FR responding in the same rats before they were trained on the PR. A more sensitive test of this question would be to look at a correlation between FR responding and PR responding in the same animals. I would encourage the authors to look into this.

2) The paper proposes to find a mechanism underlying variance in motivational drive on the PR task. As they note in their Introduction, motivation can fluctuate between days. The rats here were tested on PR until responding was stable, so the authors have data on whether these differences in PR responding were indeed stable within animals across days. They currently present PR data as the average of the last 3 days of PR training. Can they perform correlations among these days (i.e.: Days 1 vs. 2, 1 vs. 3, 2 vs. 3) to confirm that high responding on the PR task is a stable behavioral trait across days?

3) I was unclear how rats were parsed for RNA analysis, FSCV, and ephys experiments. Were these all from the original group in Figure 1A? Were these solely the lowS and highS animals identified in Figure 1A? Can the authors clarify how they divided these animals up and chose which individuals were used for each assay? A schematic of the overall experimental flowchart would be helpful for wrapping my head around it, as there were many rats that ended up in several different experiments.

4) For the ephys, can the authors clearly label how many cells from how many rats were recorded for each experiment?

5) The final section of the Results is titled, "K_v_1.4 blockade rescues deficient PR performance in lowS animals". The word "rescue" also appears in their impact statement, but I think this word is too strong considering their highS rats have breakpoints of ~250, and their lowS rats have breakpoints of ~50 which were increased to ~80 with K_v_1.4 blockade. Even with K_v_1.4 blockade these rats would be in the "lowS" group based on the distribution in Figure 1A. The authors should be more nuanced in their description of the effect as a "rescue".

[Editors' note: further revisions were requested prior to acceptance, as described below.]

Thank you for resubmitting your work entitled "Altered gating of K_v_1.4 in the nucleus accumbens suppresses motivation for reward" for further consideration at *eLife*. Your revised article has been favorably evaluated by Ronald Calabrese as the Senior Editor and three reviewers, one of whom is a member of our Board of Reviewing Editors.

The manuscript has been improved but there are two small remaining issues that need to be addressed before acceptance, as outlined below:

First, please address the concerns about statistical reporting noted by reviewer #1.

Second, given that there has been some author changes (Srimal Samaranayake and Parastoo Hashemi removed and Robert Cole added), please confirm in your cover letter that all authors, including those removed, are aware of and have approved of these changes.

Reviewer #1:

The authors have addressed most of my concerns and the manuscript is improved. I have only one remaining issue which is that there are still a few instances where statistics need to be clarified. There are still instances where only a p value is shown without the full statistical reporting, e.g., in the second paragraph of the subsection “Individual differences in motivation for sucrose reward”, all of the text for Figure 4—figure supplement 1. The full statistic needs to be shown in all these cases. In many cases, e.g. in the subsection “Low motivation for sucrose is linked to increased NAc spike output”, a two -way ANOVA is used to analyze the data, but only one F statistic is reported, leaving it in some cases unclear whether this F statistic is one of the main effects or the interaction. In these cases, please include the full ANOVA results (both main effects and interaction), or otherwise clarify which part of the ANOVA is being reported. I would be preferable to report actual p values in all cases rather than p<0.05 or p<0.01.

Reviewer #2:

The authors have appropriately responded to all of my comments in their revised manuscript. I recommend that the paper should be accepted.

*Reviewer #3:*

The authors have substantially modified their manuscript and have fully addressed my concerns. I support publication of this manuscript in its present form.

---

## [Author Response]

Essential revisions:1) The sucrose preference test data are underpowered and, moreover, these data do not reflect the hedonic experience of the pellets earned during the progressive ratio task. For that reason, the reviewers suggested removing these data. All reviewers agreed that these data were not critical, and were even tangential to the main point of this study. If they are to be kept in, power should be added and their interpretation shifted to avoid the implication that these data speak to the hedonic experience of the reward in the PR task.

Sucrose preference data have been removed.

2) The experiments assessing the effects of UK-78,282 on IA inactivation (Figure 6E) are also underpowered. Because this is such a vital component of the study, the reviewers would like to see the N for both highS and lowS subjects increased.

We have increased the animal and recorded cell numbers for the effects of UK-78,282 on I_A_ characterization for both highS and lowS subjects. These data are now based on 13-18 cells from 6 animals in each group and presented in the revised Figure 5D-E.

3) Reviewers were also concerned with the power in the FSCV and FSCAV experiments. For the FSCAV data this was especially a problem with the marginal and thus unclear effect. They agreed that these experiments are also not critical to the conclusions drawn here and can be removed. If you choose to keep them in, power should be added so both the evoked FSCV and tonic FSCAV results are more convincing.

We have removed the FSCV and FSCAV experiments from the manuscript.

4) The reviewers agreed that is not possible to adequately differentiate D1 and D2 MSNs by electrophysiological criteria alone. Thus, the claims in the Discussion, which were not supported with data, of no correlation between putative D1 or D2 identity and electrophysiological responses should be removed and instead the Discussion should make clear to the reader that this remains an open question.

We agree with the reviewers that D1/D2 identity cannot be unequivocally determined by electrophysiological criteria alone. We have extensively revised the Discussion and now state explicitly that relative contribution of D1 vs. D2 cells to our data is not known. Subsection “Spike frequency and motivation for reward”, second paragraph.

5) Because motivation in the PR task is interpreted as a stable trait-like feature, the reviewers agreed that it would be helpful provide data on the stability of PR performance across days.

This is an important point and we have revised Figure 1 to highlight stability of behavioral performance over the last 3 days of progressive ratio testing in individual lowS and highS animals. Figure 1D, E.

6) Please clarify how lowS, midS, and highS groups were defined for individual experiments and report the behavioral data for each smaller cohort.

The subjects for this study were obtained from seven different cohorts. lowS and highS groups were defined by the top and bottom quartile of the PR breakpoint interquartile distribution within each cohort. This is now illustrated in a new Figure 1—figure supplement 1A. The individual cohorts that provided subjects for RNA-seq, electrophysiology and infusion experiments are reported in the figure legend.

Reviewer #1:[…] 1) There is substantial variability in PR performance, leading to categorization of low, and high S groups for subsequent experiments, which seems reasonable. But it is not clear within these smaller cohorts used for the RNA-seq, FSCV, physiology, and infusion experiments how low, mid, and high were defined. Was this top and bottom 25% within each smaller cohort, or within the larger group? This should be made clear and the behavioral data for each smaller cohort should be shown, perhaps in a supplementary figure.

Please see essential revision #6.

2) Because motivation in the PR task is interpreted as a stable trait-like feature, it would be helpful to show that it was consistent within subject across days.

Please see essential revision #5.

3) To say that K_v_1.4 blockade 'rescues' a deficit in lowS animals is a bit of an overstatement, because even with the blockade breakpoint, presses, and earned pellets remain lower in the lowS than the midS group. To make this clear, the midS data should be included in the main Figure 7. Moreover, to make an interpretation that K_v_1.4 blockade had an effect in one group but not another, the data should be analyzed together in a 2 way ANOVA with variables group (low, mid, high S) and drug dose. A significant interaction would justify the individual group ANOVAs evaluating dose within each group.

We agree with these excellent points and addressed them in the revised manuscript as follows: 1) we refrain from using ‘rescue’ to describe the results of K_v_1.4 blockade findings. Results subsection “K_v_1.4 blockade improves PR performance in lowS animals”: 2) we included midS data in the revised Figure 6B.3)The effect of UK-78,282 across lowS, midS, and highS groups was analyzed by two-way ANOVA, which revealed significant effects of group and drug treatment, and significant interaction. These analyses have been added to Results subsection “K_v_1.4 blockade improves PR performance in lowS animals”.

4) The FSCV methods are not adequately described. The frequency of MFB electrical stimulation needs to be provided. There should be some description of how electrode placements were optimized. The FSCAV methods are not provided. Details on how the FSCV data were analyzed, including how the max was determined, how the half-life was calculated, and how transient events were defined and calculated, need to be provided.

FSCV and FSCAV experiments have been removed from the manuscript. Please see essential revision #3.

5) The 'hedonic preference' findings lead to the, perhaps unintended, implication that PR performance and reward hedonic experience are not related. But sucrose preference in this two bottle test is not reflective of the hedonic preference of the pellets earned in the PR task and so this interpretation could be somewhat misleading. I suggest referring to this as sucrose preference, as is more common, and walking back conclusions that PR and hedonia were unrelated.

We have removed results and interpretation of the sucrose preference experiments. Please see essential revision #1.

6) The N for the FSCV data are somewhat low in the lowS group (N=5), which becomes a problem for the marginal tonic DA effect. I'm concerned the data are not sufficiently powered to detect an effect. If these data are going to be shown, power should be added or a power analysis provided to indicate these experiments were sufficiently powered.

FSCV and FSCAV experiments have been removed from the manuscript. Please see essential revision #3.

Reviewer #2:[…] Overall, the study is very interesting, well conducted and of high relevance. The approach used to isolate potential underpinnings for behavioral variability is at the forefront of the field. The core of the presented data is quite adequate, and the findings are truly exciting. However, a few key issues need to be addressed.One major concern is that the sucrose preference test is woefully underpowered. It seems as if only 3 highS and 3 or 4 lowS rats were included in this test, and there is a clear trend towards the lowS animals having a higher sucrose preference. This is particularly important given the role of specific subregions of the NAc shell in mediating hedonic preference vs. incentive salience (as established in the work of Kent Berridge and others). The authors unfortunately would need to repeat this test, specifically with more highS and lowS rats, in order to have a conclusive result and rule out potential differences in sucrose preference.

We have removed results and interpretation of the sucrose preference experiments. Please see essential revision #1.

The authors also claim to have clustered their recorded cells into putative D1- and D2-expressing SPNs based on their electrophysiological properties and saw no difference in "electrophysiological responses at baseline, or following drug application" between the groups, but this data is not shown. This analysis and its results must be disclosed, as well as the relative number of putative D1 and D2 cells in each experiment, wherever this is possible. Any differential effect between these two cells types, as well as potential sampling biases, could lead to major changes in the interpretation of the results.

We excluded putative D1/D2 data clustering results from the revised Discussion, but indicate reasons why we consider sampling bias unlikely. Please see essential revision #4 and Discussion subsection “Spike frequency and motivation for reward”.

Results reported in Figure 6E are of particular concern, given that the selective effect of UK-78,282 on IA inactivation in the lowS group (a key finding of the study) is "demonstrated" with less than 6 cells in each group, which is already a relatively underpowered experiment without considering a potential asymmetry in the sampling of D1 vs. D2 subpopulations.

We have increased the number of cells and animals in electrophysiological experiments with UK78,282 and included them in the revised Figure 5D-E. Please see essential revision #2.

Reviewer #3:[…] Overall, I found the study well designed and experimentally sound, and the discussion of the questions and implications were both interesting and appropriate. I have only suggestions for improvement.1) In Figure 1, they show that the differences in PR were not observable in the preceding FR sessions. They argue this by looking at average FR responding in the same rats before they were trained on the PR. A more sensitive test of this question would be to look at a correlation between FR responding and PR responding in the same animals. I would encourage the authors to look into this.

Thank you for the idea. The revised manuscript includes correlation between FR and PR responding in individual animals in Figure 1—figure supplement 1B and Results subsection “Individual differences in motivation for sucrose reward”, second paragraph.

2) The paper proposes to find a mechanism underlying variance in motivational drive on the PR task. As they note in their Introduction, motivation can fluctuate between days. The rats here were tested on PR until responding was stable, so the authors have data on whether these differences in PR responding were indeed stable within animals across days. They currently present PR data as the average of the last 3 days of PR training. Can they perform correlations among these days (i.e.: Days 1 vs. 2, 1 vs. 3, 2 vs. 3) to confirm that high responding on the PR task is a stable behavioral trait across days?

We now include data demonstrating behavioral stability of PR over the last three days of PR training for each lowS and highS animal in Figures 1D and E.

3) I was unclear how rats were parsed for RNA analysis, FSCV, and ephys experiments. Were these all from the original group in Figure 1A? Were these solely the lowS and highS animals identified in Figure 1A? Can the authors clarify how they divided these animals up and chose which individuals were used for each assay? A schematic of the overall experimental flowchart would be helpful for wrapping my head around it, as there were many rats that ended up in several different experiments.

Please see essential revision #6.

4) For the ephys, can the authors clearly label how many cells from how many rats were recorded for each experiment?

Cell and animal number information has been added throughout the manuscript.

5) The final section of the Results is titled, "K_v_1.4 blockade rescues deficient PR performance in lowS animals". The word "rescue" also appears in their impact statement, but I think this word is too strong considering their highS rats have breakpoints of ~250, and their lowS rats have breakpoints of ~50 which were increased to ~80 with K_v_1.4 blockade. Even with K_v_1.4 blockade these rats would be in the "lowS" group based on the distribution in Figure 1A. The authors should be more nuanced in their description of the effect as a "rescue".

We changed the title of the final section of the Results to “K_v_1.4 blockade improved PR performance in lowS animals” and refrain from unwarranted use of “rescue” in the revised impact statement and throughout the revised manuscript.

[Editors' note: further revisions were requested prior to acceptance, as described below.]The manuscript has been improved but there are two small remaining issues that need to be addressed before acceptance, as outlined below:First, please address the concerns about statistical reporting noted by reviewer #1.Second, given that there has been some author changes (Srimal Samaranayake and Parastoo Hashemi removed and Robert Cole added), please confirm in your cover letter that all authors, including those removed, are aware of and have approved of these changes.

In this revised version, we addressed the statistical reporting comments raised by reviewer #1.

We have also removed two authors: Srimal Samaranayake and Parastoo Hashemi. These two authors provided fast-scan cyclic voltammetry data for the original submission. The voltammetry data have been removed from the revised version per reviewers’ suggestion. Both Dr. Samaranayake and Dr. Pashemi are aware that their names have been removed from the author list and approved this change. Additionally, in the revised version, Robert Cole was added to the author list. Dr. Cole was involved in experiments essential for the timely completion of the revised work. Dr. Cole is aware of and approves his addition to the author list of the revised manuscript.